# Expression of Mismatch Repair Proteins in Merkel Cell Carcinoma

**DOI:** 10.3390/cancers13112524

**Published:** 2021-05-21

**Authors:** Thilo Gambichler, Nessr Abu Rached, Andrea Tannapfel, Jürgen C. Becker, Markus Vogt, Marina Skrygan, Ulrike Wieland, Steffi Silling, Laura Susok, Markus Stücker, Thomas Meyer, Eggert Stockfleth, Klaus Junker, Heiko U. Käfferlein, Thomas Brüning, Kerstin Lang

**Affiliations:** 1Skin Cancer Center, Department of Dermatology, Ruhr-University Bochum, 44791 Bochum, Germany; m.skrygan@klinikum-bochum.de (M.S.); laura.susok@klinikum-bochum.de (L.S.); markus.stuecker@klinikum-bochum.de (M.S.); thomas.meyer@klinikum-bochum.de (T.M.); eggert.stockfleth@klinikum-bochum.de (E.S.); 2Institute of Pathology, Ruhr-University Bochum, 44789 Bochum, Germany; andrea.tannapfel@pathologie-bochum.de (A.T.); markus.vogt@pathologie-bochum.de (M.V.); 3Translational Skin Cancer Research, German Cancer Consortium (DKTK) Partner Site Essen/Düsseldorf, Department of Dermatology, University Duisburg-Essen, 45147 Essen, Germany; j.becker@dkfz-heidelberg.de; 4Deutsches Krebsforschungszentrum (DKFZ), 69120 Heidelberg, Germany; 5National Reference Center for Papilloma- and Polyomaviruses, Institute of Virology, University of Cologne, 50935 Cologne, Germany; ulrike.wieland@uni-koeln.de (U.W.); steffi.silling@uk-koeln.de (S.S.); 6Department of Pathology, Klinikum Bremen-Mitte, 28205 Bremen, Germany; klaus.junker@klinikum-bremen-mitte.de; 7Institute for Prevention and Occupational Medicine of the German Social Accident Insurances (IPA), Ruhr-University Bochum, 44789 Bochum, Germany; kaefferlein@ipa-dguv.de (H.U.K.); bruening@ipa-dguv.de (T.B.); lang@ipa-dguv.de (K.L.)

**Keywords:** Merkel cell carcinoma, Merkel cell polyomavirus, mismatch repair deficiency, microsatellite instability, immune checkpoint inhibitors, immunotherapy

## Abstract

**Simple Summary:**

Merkel cell carcinoma (MCC) is a rare and highly malignant skin cancer with neuroendocrine differentiation. About 80% are Merkel cell polyomavirus (MCPyV) positive. The aim of this work was to immunohistochemically investigate the expression of mismatch repair proteins (MSH2, MSH6, MLH1, and PMS2) in MCC (*n* = 56). In a second step, tumors with a low expression were tested for microsatellite instability. Microsatellite instability in MCC could have an impact on immune checkpoint inhibitor therapy (ICI) outcome. This study showed a significant association between low expression of mismatch repair proteins and a negative MCPyV status. Microsatellite instability was detected in only one case. Future studies will establish whether this subset of MCC patients respond better to ICI.

**Abstract:**

We aimed to assess for the first time the mismatch repair (MMR) protein expression in Merkel cell carcinoma (MCC). Immunohistochemistry was performed for MLH1, MSH2, MSH6, and PMS2 on patients’ tumor tissue (*n* = 56), including neighbored healthy control tissue. In cases with low-level MMR expression (<10th percentile), we performed multiplex PCR in combination with high-resolution capillary electrophoresis in order to confirm microsatellite instability (MSI). Microscopic evaluation revealed a high median expression for all MMR proteins studied (91.6–96.3%). However, six patients (56/10.7%) had low-level MLH1 expression, six (55/10.9%) had low-level MSH2 expression, five (56/8.9%) had low-level MSH6 expression, and six (54/11.1%) had low-level PMS2 expression. Together, we observed nine (56/16.1%) patients who had low-level MMR expression of at least one protein. Of the patients with low-level MMR expression, MSI evaluation was possible in five cases, revealing one case with high-level MSI. In all MMR proteins assessed, low-level expression was significantly (*p* = 0.0004 to *p* < 0.0001) associated with a negative Merkel cell polyomavirus (MCPyV) status. However, the expression profiles of the MMR proteins did not correlate with clinical outcome measures such as disease relapse or death (*p* > 0.05). MCC appears to be a malignancy characterized by low-level MMR rather than completely deficient MMR in a subset of cases, predominantly affecting MCPyV-negative tumors. Future studies will establish whether this subset of MCC patients respond better to immune checkpoint inhibitor therapy.

## 1. Introduction

Merkel cell carcinoma (MCC) is a highly aggressive skin cancer that is typically cytokeratin 20 positive on immunohistochemistry. Merkel cell polyomavirus (MCPyV) is clonally integrated in the majority of MCC in patients of the Northern Hemisphere [1]. The incidence of MCC is currently about 0.4/100,000 cases per year. A high local recurrence rate, regional lymph node metastases, and distant metastases are typical biological characteristics of MCC. Major risk factors for MCC are chronic UV exposure, high age, and immune suppression [1,2,3,4,5]. When compared to previous chemotherapeutic modalities, management of advanced MCC has significantly been improved since the introduction of immune checkpoint inhibitors (ICI) such as anti-programmed cell death protein 1 (PD-1) inhibitors (pembrolizumab, nivolumab) and anti-programmed cell death ligand protein 1 (PD-L1) inhibitors (avelumab) [6,7,8]. ICI in the metastatic setting of MCC are frequently associated with durable response rates (≈70%) and the 3-year overall survival of about 65% [8,9]. However, there is a lack of knowledge regarding the molecular predictors of ICI response in MCC. With respect to tumor cell characteristics, Kacew et al. [9] reported that single-nucleotide variants in *ARID2* and *NTRK1* genes are associated with response to ICI, whereas the MCPyV status, total mutational burden (TMB), UV mutational signatures, and copy-number alterations did not correlate with treatment response [8,9]. In many cancers, including colorectal, endometrial, prostate, and bladder cancer, small satellite DNA damage results in microsatellite instability (MSI) and consecutive mismatch repair (MMR) deficiency, which may have prognostic consequences. For instance, the prognosis of most malignancies with high-level MSI (MSI-H) and deficient MMR (dMMR) is good, in particular when treated with ICI [10]. Interestingly, ICI also have been demonstrated to be more effective in high-grade neuroendocrine tumors with high TMB, MSI, and/or mutational load [11,12]. So far, no papers have been published on MMR protein expression and MSI in MCC.

The main aim of this study was to determine for the first time the expression profiles of MMR proteins and search for MSI-H in selected cases with low-level MMR or dMMR.

## 2. Materials and Methods

### 2.1. Patients

Diagnosis of MCC has been verified by two experienced dermato-histopathologists according to valid histopathology and immunohistology criteria. The current national guidelines were used for the diagnostics, clinical work-up, and follow-up [2,4]. MCC restaging was performed in accordance with the 8th edition of the AJCC guidelines [4,13]. Missing clinical data were completed by means of chart review as well as contacting patients, relatives, general practitioners, and dermatologists. The study was approved by the local ethics review board of the Medical Faculty of the Ruhr-University Bochum (#4749-13).

### 2.2. Analysis of Human Polyomavirus in Formalin-Fixed, Paraffin-Embedded (FFPE) Tissue

Using a LightCycler 480 Real Time PCR System (Roche, Grenzach, Germany), the MCPyV viral load was assessed, as previously reported by Wieland et al. [14]. In brief, the load of MCPyV was measured by means of quantitative RT-PCR (Roche, Grenzach, Germany) using MCPyV-specific LT3-primers as well as a locked nucleic acid probe that binds to the N-terminal locus of the large T-antigen gene [2]. The DNA load of MCPyV was given in MCPyV DNA copies/betaglobin-gene copies [15].

### 2.3. Immunohistochemistry of MCC Tumor Samples

Staining for MLH1, MSH2, MSH6, and PMS2 was performed as follows: 4 µm sections from FFPE blocks were mounted on DAKO IHC Microscope Slides (Agilent, Hamburg, Germany) and stored for 30 min at 56 °C. Sections were deparaffinized in Rotihistol (Carl Roth, Karlsruhe, Germany) (10 min, RT, 2 times) and subsequently hydrated through a graded alcohol series. Antigen retrieval was performed by cooking sections for 20 min in an EnVision Flex target retrieval solution (K8004; Agilent, Hamburg, Germany), ‘High pH’, in a steamer. Blocking of unspecific staining was accomplished by using Dako Dual Endogenous Enzyme Block (S2003; Agilent, Hamburg, Germany) (15 min, RT), and additionally 1.5% casein for PMS2 (M3647; Agilent DAKO, Hamburg, Germany) (15 min, RT). For immunostaining, we used rabbit monoclonal antibodies against PMS2 and MSH6 (M3646; Agilent DAKO, Hamburg, Germany), and mouse monoclonal antibodies against MLH1 (M3640; Agilent DAKO, Hamburg, Germany) and MSH2 (M3639; Agilent DAKO, Hamburg, Germany). All antibodies were derived from Agilent. The diluted antibodies against MLH1 (1:50), MSH2 (1:50), and MSH6 (1:50) were incubated for 20 min and against PMS2 (1:40) for 30 min at room temperature in a humidified chamber. As the negative control, sections were incubated without using a primary antibody. The antigen was stained red by the use of the Dako REAL^TM^ Detection System, Alkaline Phosphatase/RED, Rabbit/Mouse (K5005; Agilent DAKO, Hamburg, Germany) in accordance with the manufacturer’s recommendations, and blue with hematoxylin for nuclear counterstaining. Finally, samples went through a series of ascending alcohol concentrations and were mounted with Entellan (Merck, Darmstadt, Germany).

### 2.4. Microscopic Evaluation

Microscopic evaluation was carried out using the stained specimens that were previously scanned at twentyfold magnification with the aid of the Nanozoomer Whole Slide Scanner (Hamamatsu, Herrsching am Ammersee, Germany). All scanned slides were assessed by means of the NDP.view2 software (Hamamatsu Photonics, Hamamatsu City, Japan). Program Count Helper (Massako Sakanashi, Sakanapps, Japan) was used to help with manual cell counting. All tumor cells on the entire slide were evaluated with respect to the nuclear staining of each protein. Protein expression was expressed as the % of nuclear-stained tumor cells relative to all tumor cells on the slide. In accordance with the College of American Pathologists guidelines for immunohistology evaluation [16,17], any nuclear tumor cell staining (even patchy) was taken as “no loss of expression” and only complete absence of nuclear staining was considered “loss of expression” provided that internal controls (e.g., keratinocytes, lymphocytes, and stromal cells) showed staining. Hence, MMR deficiency was considered when there was complete absence of nuclear staining for at least one protein. Cases with an MMR protein expression of less than the 10th percentile were classified as low-level MMR, and cases with an expression of more than the 10th percentile as high-level MMR.

### 2.5. Multiplex-PCR and HRCE

Following microdissection of the FFPE material, we extracted DNA from tumorous and neighbored non-tumorous tissue. We assessed changes in fragment lengths both for mono-nucleotide and di-nucleotide markers (BAT25, BAT26 D2S123, D5S346, and D17S250, respectively) by means of multiplex PCR combined with high-resolution capillary electrophoresis (HRCE). MSI-H was defined if ≥2 out of 5 markers were found to be instable.

### 2.6. Statistics

Data analysis was performed using the statistical package MedCalc Software version 19.1.7. (MedCalc Software, Ostend, Belgium). Distribution of data was assessed by the D‘Agostino–Pearson test. Normally distributed data were expressed as the mean and standard deviation (SD), and non-normally distributed data as the medians and range. Where appropriate, data were analyzed using the Mann–Whitney test, Spearman correlation procedures, and Chi^2^ test. *p*-values < 0.05 were considered significant.

## 3. Results

### 3.1. Patients’ Characteristics

Fifty-six patients (median age: 77.5 years (51–95); 27 males, 29 females)) were investigated, including 11/56 (19.6%) patients with MCPyV-negative and 45/56 (80.4%) MCPyV-positive MCC. For all of them, their formalin-fixed, paraffin-embedded (FFPE) tumor tissue was available. At the time of diagnosis, including the first complete work-up, 21 patients (56; 37.5%) were in stage I, 19 (56; 33.9%) in stage IIA, 1 (56; 1.8%) in stage IIB, 4 (56; 7.1%) in stage IIIA, 6 (56; 10.7%) in stage IIIB, and 5 (56; 8.9%) in stage IV, according to the 8th edition of the American Joint Committee on Cancer staging system for MCC [13]. In total, 24 (42.9%) out of 56 primary tumors were observed in high-risk regions (head/neck) and 12 (56/21.4%) patients were immunosuppressed.

### 3.2. Expression of MMR Proteins in MCC

Microscopic evaluation revealed high median (range) expression for all MMR proteins studied (Table 1): MLH1 96.3% (8.3–99.8), 10th percentile 58%; MSH2 94.7% (7.2–99.6), 10th percentile 74%; MSH6 91.6 % (16.2–99.1), 10th percentile 52%; PMS2 93.1 (6.4–99.3), 10th percentile 32%. Six patients (56/10.7%) had low-level MLH1 expression, six (55/10.9%) had low-level MSH2 expression, five (56/8.9%) low-level MSH6 expression, and six (54/11.1%) low-level PMS2 expression (Figure 1). Together, we observed nine (56/16.1%) patients who had low-level MMR expression of at least one protein. Hence, none of the MMR proteins assessed showed complete absence of immunoreactivity and thus no dMMR per definition. In all MMR proteins assessed, low-level expression was significantly (r = 0.48 to 0.70; *p* = 0.0004 to *p* < 0.0001) associated with a negative MCPyV status. MCPyV load negatively correlated (r = −0.39, *p* = 0.039) with tumor cell proliferation evaluated by Ki-67 staining. The latter did not correlate with MMR expression (*p* > 0.05). The expression profiles of MMR proteins highly and significantly correlated with each other (r = 0.51 to r = 0.60; *p* = 0.0001 to *p* < 0.0001). However, the expression profiles of the MMR proteins did not correlate with clinical outcome measures such as disease relapse or death (*p* > 0.05).

### 3.3. Results of MSI Testing

Of the nine patients with low-level MMR protein expression, MSI evaluation was possible in five cases, revealing four cases of being microsatellite stable (MSS) and one patient with MSI-H. The latter was a patient with low-level MLH1 (8.3%) and PMS2 (6.4%) expression. In four cases, MSI testing was not possible due to technical reasons or missing tumor tissue.

### 3.4. Patients’ Treatment and Outcome

The patients were managed in line with the German guideline for MCC [4]. Accordingly, all primaries were completely removed including a safety margin of one to two cm. Following sentinel lymph node biopsy, the patients were treated with adjuvant radiotherapy for the tumor bed and draining lymph node basin. Metastatic lymph node disease was treated with complete lymphadenectomy. Most patients of advanced stage received radiotherapy, electrochemotherapy, and systemic chemotherapy (e.g., carboplatin, etoposide). Within a median progression-free survival period of 12 months (2–60 months) 22 patients (56; 39.3%) experienced a disease recurrence, and 18 (56; 32.1%) patients died from MCC within a median follow-up period of 26 months (3–60 months; Table 1 and Figure 2). Thus, the MCC-specific death rate was 32.1%, whereas the overall survival rate was 51.8%. MCC recurrence rates did not significantly (*p* = 0.26) differ between patients with high-level MMR expression (20/47/42.5%) and patients with low-level MMR (2/9/22.2%). Moreover, progression-free survival and MCC-specific survival time did not significantly (*p* = 0.6 and *p* = 0.18, respectively) differ between high-level MMR patients (13 months and 24 months, respectively) and low-level MMR patients (6 months and 6 months, respectively).

Since most patients were treated in the pre-ICI era, only nine patients of advanced stages actually received ICI. One of these patients showed low-level expression of all MMR proteins assessed and underwent 24 cycles of avelumab for inoperable in-transit metastatic MCPyV-negative MCC. On a 40-month follow-up period, this patient did not show MCC recurrence. All other ICI-treated patients had intact MMR expression, of which three died of MCC, one from other cause, and four were alive; the latter had a very short follow-up period. The patient with MSI-H and low-level MLH1 and PMS2 expression had stage I MCPyV-negative MCC and survived at least 5 years without disease recurrence.

## 4. Discussion

The American Society for Clinical Pathology, College of American Pathologists, Association for Molecular Pathology, and the American Society of Clinical Oncology strongly recommend the evaluation of MSI/MMR biomarkers in colorectal cancer for a better prognostic stratification of patients. This recommendation is emphasized by the recent evidence of MSI as a predictive factor for response to ICI [18]. For example, the Food and Drug Administration has recently approved pembrolizumab as first-line therapy for MSI-H/dMMR metastatic colorectal cancer [19]. The prevalence of MSI-H/dMMR deficiency differs between gastrointestinal cancers. It occurs most frequently in colorectal (up to 15%) and gastric cancer (about 10%) [19], and less frequently in hepatocellular-/cholangiocarcinoma and esophageal and pancreatic adenocarcinoma (< 5%) [19]. With the dramatic response of MSI-H/dMMR-deficient tumors to ICI, MSI/MMR testing has, however, increased significantly in many solid tumors.

Notably, there exist relatively little data on MSI/MMR in cutaneous malignancies. Reuschenbach et al [20] studied MSI using BAT25, BAT26, and CAT25 markers in 141 epithelial skin lesions, including squamous cell carcinoma, Bowen’s disease, actinic keratosis, keratoacanthoma, and basal cell carcinoma. None of the 141 analyzed skin lesions displayed MSI at any of the assessed markers [20]. Based on their results and the data reported in previous studies, the authors concluded that MSI-H/dMMR is not a relevant tumorigenic mechanism in non-melanoma skin cancer [20,21,22]. By contrast, MSI-H/dMMR might be more relevant in cutaneous melanoma (CM) and its responsiveness to ICI. Kubeček and Kopecký [23] concluded that the data on MSI-H/dMMR prevalence, pathogenesis, and clinical consequences in CM are still relatively limited. Korabiowska et al. [24] suggested that in CM, a reduced expression of MMR proteins, rather than a complete loss, is of importance, as confirmed by both immunohistochemistry and in situ hybridization in 59 CMs. Alvino et al. [25] also reported a reduction in expression of MLH1, MSH2, and PMS2 in CM compared to benign nevi. Interestingly, high MSH6 expression in CM was significantly associated with an increased risk of CM mortality. Roncati [26] reported a mucosal CM patient with dMMR (exclusively for MSH6) who experienced long-term disease control using pembrolizumab. Moreover, Ponti et al. [27] studied 14 CM patients receiving anti-PD-1 therapy. They performed immunohistochemistry for MLH1, MSH2, MSH6, and PMS2 on primary tumors and several metastases. Their data showed that 7% of the primary CM tissue obtained from the patient cohort exhibited dMMR in at least one protein. Three samples from one patient, including one primary melanoma and two metastases, exhibited dMSH6 expression and had the most successful response to anti PD-1 treatment [27].

As discussed above, single-nucleotide variants in the *ARID2* gene are associated with response to ICI in MCC. Shen et al. [28] showed that loss of ARID1A leads to increased MSI with an inability to recruit MMR genes during DNA repair, thus increasing the mutational burden and neoantigen load. In the present study, we demonstrated that dMMR per definition appears to be absent in MCC. Similar to CM [25], however, we found in 16.1% of primary tumors low-level MMR across all MMR proteins studied. The observed low-level MMR significantly correlated with a negative MCPyV status. This finding is in agreement with data demonstrating that, unlike MCPyV-positive MCC, MCPyV-negative MCC is characterized by high TMB and UV mutational signatures. Low-level MMR protein expression, however, correlated with MSI-H only in one case. Inconsistencies between dMMR and MSI-H status have been reported in the literature. About 10% of MSI-H cancers are evaluated as normal by immunohistochemistry, since they have non-functional MMR proteins. Vice versa, for example, loss of the MSH6 protein on immunohistochemistry may be associated with MSS or MSI-L tumors [12]. As expected, the immunoreactivity of the four MMR proteins studied significantly correlated with each other. However, we observed no significant association between MMR expression and clinical outcome. This outcome may particularly due to the small number of patients who had received ICI. Notably, one patient with MCPyV-negative MCC and low-level MMR showed a favorable long-term outcome after ICI therapy. Nevertheless, the subset of patients who received ICI was too small to draw firm conclusions.

As reported by Mandal el al. [29], however, not all dMMR tumors show good response to ICI. In an animal model, the authors showed that the genome-wide intensity of MSI and resultant TMB affects response to ICI and tumor evolution in dMMR tumors. Mandal et al. [29] concluded that the basis for this response may probably be multifactorial and could disproportionately rely on indel mutations over missense mutations to drive clinical outcome. Altogether, MSI-H/dMMR does occur in many solid tumors and frequently represents a predictive marker for response to ICI [30]. Similar to observations in CM, however, MCC appears not to be a malignancy characterized by dMMR but by low-level MMR in a subset of cases, predominantly affecting MCPyV-negative tumors. Future studies will establish whether this subset of MCC patients respond better to ICI [31,32].

## 5. Conclusions

We have shown for the first time that MCPyV-negative MCC is a malignancy characterized by low-level MMR rather than dMMR. Future studies will establish whether this subset of MCC patients respond better to ICI.

## Figures and Tables

**Figure 1 cancers-13-02524-f001:**
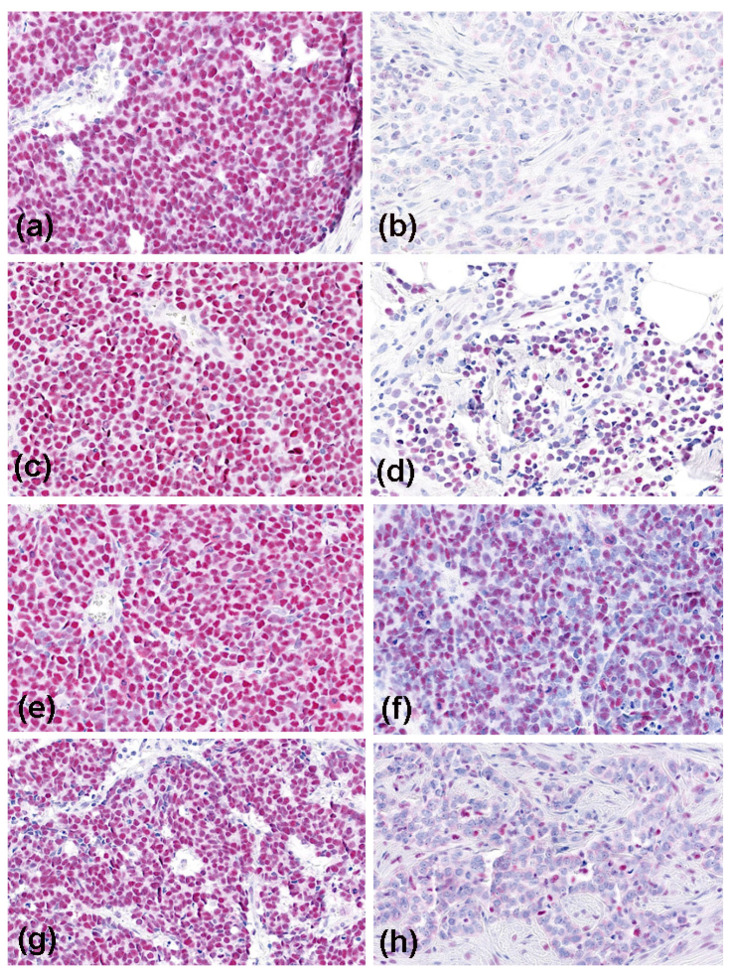
Immunoreactivity of the mismatch repair proteins in Merkel cell carcinoma (magnification, ×200). High expression (all MCPyV-positive cases) is shown on the left side: MLH1 (**a**), MSH2 (**c**), MSH6 (**e**), and PMS2 (**g**); and low-level expression (all MCPyV-negative cases) on the right side: MLH1 (**b**), MSH2 (**d**), MSH6 (**f**), and PMS2 (**h**).

**Figure 2 cancers-13-02524-f002:**
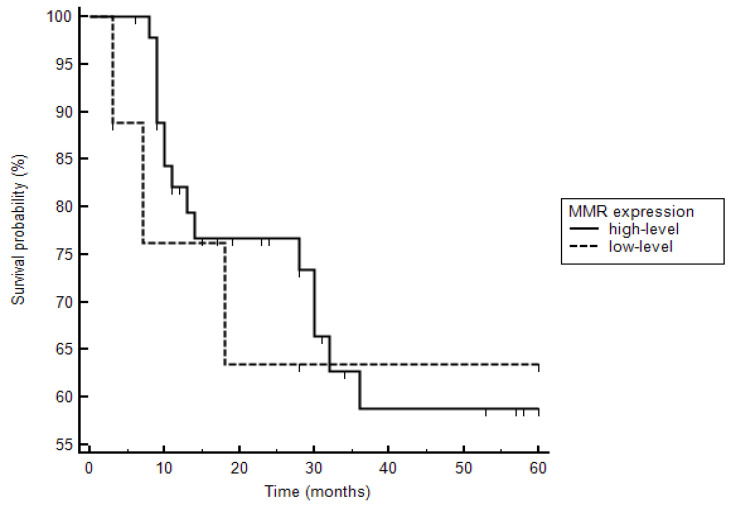
Kaplan–Meier curves are demonstrated with respect to deaths in patients with high- and low-level mismatch repair protein (MMR) expression. Three of six patients with low-level MMR expression died, whereas 15 of 47 patients with high-level MMR expression died (hazard ratio: 1.2, CI 0.31 to 4.3; *p* = 0.81).

**Table 1 cancers-13-02524-t001:** Clinical characteristics and results of the mismatch repair protein analysis in patients with Merkel cell carcinoma (MCC, *n =* 56).

Parameters	Data
Age at diagnosis * (years)	77.5 (51–95)
Gender m/f	27/29 (48.2%/51.8%)
Primary MCC localization Head/neck (no/yes) MCPyV (negative/positive)	32/24 (57.1%/42.9%) 11/45 (19.6%/80.4%)
Tumor stage at diagnosis (AJCC 2017)	I 21 (37.5%) IIA 19 (33.9%) IIB 1 (1.8%)
	IIIA 4 (7.2%) IIIB 6 (10.7%) IV 5 (8.9%)
**Mismatch repair protein expression ***	
(% positive tumor cells)	96.3% (8.3–99.8)
MLH1	58%
10th percentile	6 (56/10.7%), all MCPyV-negative
Patients with low-level **	94.7% (7.2–99.6)
MSH2	74%
10th percentile	6 (55/10.9%), all MCPyV-negative
Patients with low-level **	91.6% (16.2–99.1)
MSH6	52%
10th percentile	5 (56/8.9%), 4 MCPyV-negative
Patients with low-level **	
PMS2	93.1% (6.4–99.3)
10th percentile	32%
Patients with low-level **	6 (54/11.1%), all MCPyV-negative
Outcome	
5-year MCC relapse (no/yes)	34/22 (60.7%/39.3%)
Median time to relapse (months) *	12 (2–60)
5-year MCC (survived/deceased)	38/18 (67.9%/32.1%)
Median time to death (months) *	26 (3–60)

MCPyV = Merkel cell polyomavirus; * medians and range; ** expression <10th percentile.

## Data Availability

Derived data supporting the findings of this study are available from the corresponding author N.AR on reasonable request.

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
