# Peer review of "Expression of Mismatch Repair Proteins in Merkel Cell Carcinoma"

_cancers, 2021, doi:10.3390/cancers13112524_

Round 1
Reviewer 1 Report
This manuscript assessed the expression of mismatch repair (MMR) proteins in human Merkel cell carcinoma (MCC) tumor samples, and reported that low expression of MMR is associated with negative Merkel cell polyomavirus status in MCC, while MMR expression level has no correlation with clinical outcomes. Revisions are needed.
- In the abstract, the authors wrote that “low-level expression was significantly (P = 0.0004 to P < 0.0001) associated with negative MCC polyomavirus (MCPyV) status”, no data was shown in the figures or tables to support this conclusion. In figure 1, the authors should show representative images for staining of the 4 proteins in MCPyV positive and negative MCC samples, and show results of correlation and statistical analysis.
- The authors wrote in the methods that “Cases with an MMR protein expression of less than 10th percentile were classified as low-level MMR, and cases with an expression of ≥ 10% percentile as sufficient MMR”, however, in the text and figure 1, the authors use terms of “low-level” and “high expression” to indicate MMR expression levels. The authors should define “high expression” in the method.
- Figure 1d and 1e, it looks like more than 10% of tumor cells were nuclei-stained in those panels, which is not consistent with the definition in Methods that “expression of less than 10th percentile were classified as low-level MMR”.
- The authors summarized the clinical characteristics of MCC and results of MMR expression analysis in Table 1, however, data should be shown in the Results section to support the conclusions in Abstract that “the expression profiles of MMR proteins did not correlate with clinical 31 outcome measures such as disease relapse or death (P > 0.05)”. It would be helpful if the authors can show the correlation analysis results in figures for this conclusion even it’s not significant.
- In the literature, full name of MCPyV is “Merkel cell polyomavirus”, not “MCC polyomavirus”.
- Writing can be improved. Grammar is irregular at many points.
Author Response
Reviewer 1
This manuscript assessed the expression of mismatch repair (MMR) proteins in human Merkel cell carcinoma (MCC) tumor samples, and reported that low expression of MMR is associated with negative Merkel cell polyomavirus status in MCC, while MMR expression level has no correlation with clinical outcomes. Revisions are needed.
- In the abstract, the authors wrote that “low-level expression was significantly (P = 0.0004 to P < 0.0001) associated with negative MCC polyomavirus (MCPyV) status”, no data was shown in the figures or tables to support this conclusion. In figure 1, the authors should show representative images for staining of the 4 proteins in MCPyV positive and negative MCC samples, and show results of correlation and statistical analysis.
We have statistically analysed the association between MCPyV status and low-level MMR by means of Chi² test and Spearman procedure. We have now included the coefficients of correlation and P-values in the result section and showed in Table how many patients with low-level MMR had MCPyV-negative status. Moreover, we stated in figure legend 1 that the high-level MMR were associated with MCPyV-positive status and the low-level MMR with MCPyV-negative status.
- The authors wrote in the methods that “Cases with an MMR protein expression of less than 10th percentile were classified as low level MMR, and cases with an expression of ≥ 10% percentile as sufficient MMR”, however, in the text and figure 1, the authors use terms of “low-level” and “high expression” to indicate MMR expression levels. The authors should define “high expression” in the method.
As requested we have defined high-level as ≥ 10th percentile expression in the method section.
- Figure 1d and 1e, it looks like more than 10% of tumor cells were nuclei-stained in those panels, which is not consistent with the definition in Methods that “expression of less than 10th percentile were classified as low-level MMR”.
The actual percentage of MMR expression was given in Tab. 1. The cut-off level was 32 - 74 percent (being the 10th percentile) of positive cells. What we meant in the result section that about 10% of patients had low-level MMR.
- The authors summarized the clinical characteristics of MCC and results of MMR expression analysis in Table 1, however, data should be shown in the Results section to support the conclusions in Abstract that “the expression profiles of MMR proteins did not correlate with clinical 31 outcome measures such as disease relapse or death (P > 0.05)”. It would be helpful if the authors can show the correlation analysis results in figures for this conclusion even it’s not significant.
We have added a new figure showing Kaplan-Meier curves including the hazard ratios and P-values for each MMR protein.
- In the literature, full name of MCPyV is “Merkel cell polyomavirus”, not “MCC polyomavirus”.
We did not find the term “MCC polyomavirus” in the text!?
- Writing can be improved. Grammar is irregular at many points.
We have revised the manuscript accordingly.
Reviewer 2 Report
Well written article.
The authors discuss about patient follow up, however key presentation of the data is lacking. A Kaplan-Meier survival plot and graphs needs to be included for proper and clarified presentation of the data is essential.
Thanks.
Author Response
Reviewer 2 2 Well written article.The authors discuss about patient follow up, however key presentation of the data is lacking. A Kaplan-Meier survival plot and graphs needs to be included for proper and clarified presentation of the data is essential.
Round 2
Reviewer 1 Report
Questions have been addressed, I don't have further comments
Author Response
1) Questions have been addressed, I don't have further comments
Thank you for the detailed feedback.